# Tactile Sensing and its Role in Learning and Deploying Robotic Grasping Controllers

Alexander Koenig[1,2], Zixi Liu[2], Lucas Janson[3] and Robert Howe[2,4]

*Abstract*— A long-standing question in robot hand design is how accurate tactile sensing must be. This paper uses simulated tactile signals and the reinforcement learning (RL) framework to study the sensing needs in grasping systems. Our first experiment investigates the need for rich tactile sensing in the rewards of RL-based grasp refinement algorithms for multi-fingered robotic hands. We systematically integrate different levels of tactile data into the rewards using analytic grasp stability metrics. We find that combining information on contact positions, normals, and forces in the reward yields the highest average success rates of 95.4% for cuboids, 93.1% for cylinders, and 62.3% for spheres across wrist position errors between 0 and 7 centimeters and rotational errors between 0 and 14 degrees. This contact-based reward outperforms a non-tactile binary-reward baseline by 42.9%. Our follow-up experiment shows that when training with tactile-enabled rewards, the use of tactile information in the control policy's state vector is drastically reducible at only a slight performance decrease of at most 6.6% for no tactile sensing in the state. Since policies do not require access to the reward signal at test time, our work implies that models trained on tactile-enabled hands are deployable to robotic hands with a smaller sensor suite, potentially reducing cost dramatically.

## I. INTRODUCTION

Tactile sensing provides information about local object geometry, surface properties, contact forces, and grasp stability [1]. Hence, tactile sensors can be a valuable tool in contact-rich scenarios such as robotic grasp refinement [2] where a grasping system recovers from calibration errors. Computer vision approaches for grasp refinement often face limitations due to the occlusion of contact events. Tactile sensors can be expensive and fragile hardware components. Hence, for cost-effective robotic hand design, it is essential to understand when robot hands need precise sensing and how accurate it should be to achieve good grasping performance.

A few research papers investigated the effect of tactile sensor resolution on grasp success. Wan et al. [3] found that reduced spatial resolution of tactile sensors negatively impacts grasp success since inaccuracies in contact position and normal sensing can influence grasp stability predictions. Other works analyzed the effect of contact sensor resolution on grasp performance in the context of reinforcement learning. In simulated experiments, Merzić et al. [4] found

This material is based upon work supported by the US National Science Foundation under Grant No. IIS-1924984 and by the German Academic Exchange Service. An extended paper including the material in this abstract has been submitted for publication.

[1] Department of Informatics, Technical University of Munich
[2] School of Engineering and Applied Sciences, Harvard University
[3] Department of Statistics, Harvard University
[4] RightHand Robotics, Inc., 237 Washington St, Somerville, MA 02143 USA. Robert Howe is corresponding author howe@seas.harvard.edu.

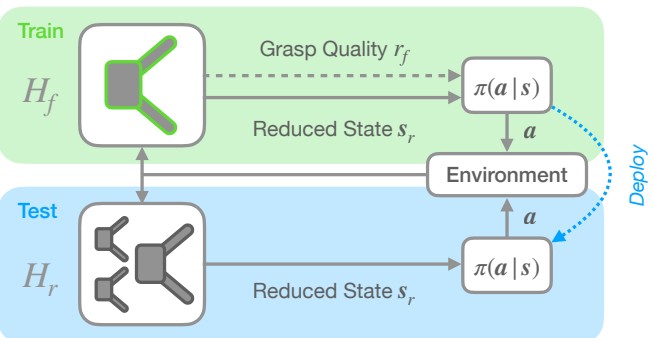

Fig. 1: The hypothesized workflow for training and deploying RL-controlled grasping systems. First, train a policy $\pi(a|s)$ on a hand $H_f$ with a *full* tactile sensor suite (e.g., contact position, normal and force sensors) where the grasp quality metrics are available as a reward $r_f$ to learn a task, but only provide a subset of the available contact data in the state vector $s_r$. Afterwards, deploy the policy to many structurally similar hands $H_r$ with a *reduced* sensor set to save cost.

that contact feedback in a policy's state vector improves the performance of RL-based grasping controllers, and [5], [6] presented similar results for in-hand manipulation. However, [5], [6] also concluded that models trained with binary contact signals perform equally well as models that receive accurate normal force information. Furthermore, [5], [6] found that tactile resolution (92 vs. 16 sensors) has no noticeable effect on performance and sample efficiency of reinforcement learned manipulation controllers.

In this paper, we use accurate tactile signals from simulation and the reinforcement learning framework to explore the tactile sensing needs in robotic systems. RL algorithms aim to produce a policy $\pi(a|s)$ that outputs actions $a$ given state information $s$ such that the cumulative reward signal $r$ is maximized. The reward function is a critical part of every RL algorithm [7]. While the previous work in [4], [5], [6] only studied the tactile resolution in the policy's state, our first contribution investigates the impact of tactile information in the reward signal. We propose a unified framework to systematically incorporate different levels of tactile information from robotic hands into a reward signal via analytic grasp stability metrics. We conduct grasp refinement experiments on two types of quality metrics discussed in Section II: $\epsilon$ [8] calculated from contact positions and normals and a contact force-based reward $\delta$. In Section III, we estimate the relevance of contact position, normal, and force sensing for the reward signal by comparing the individual and combined performance of $\epsilon$ and $\delta$.

Calculating grasp stability metrics requires costly tactile sensing capabilities on physical grippers. However, the reward signal is only required during the training of policies but not while testing, which suggests that sensing needs in both stages could be different. We hypothesize in Fig. 1 that policies trained with grasp stability metrics on a robotic hand $H_f$ with a *full* tactile sensor suite are deployable to structurally similar but more affordable hands $H_r$ with *reduced* tactile sensing at a small performance decrease. Hence, our second experiment in Section IV gradually decreases tactile resolution in the state vector to find realistic training and deployment workflows for grasping algorithms.

## II. GRASP STABILITY METRICS

### A. Largest-minimum resisted forces and torques

Mirtich and Canny [8] define two quality metrics $\epsilon_f$ and $\epsilon_\tau$ that measure a grasp's ability to resist unit forces and torques, respectively. As discussed in [9], the friction cone constrains the contact force $\boldsymbol{f}_i$ at each contact $i$. It is discretized using $m$ edges $\boldsymbol{f}_{i,j}$. The set of forces $\mathcal{W}_f$ that the contacts can apply to the object is $\mathcal{W}_f = \mathrm{ConvexHull}\left(\bigcup_{i=1}^{n_c}\{\boldsymbol{f}_{i,1},\ldots,\boldsymbol{f}_{i,m}\}\right)$, where $n_c$ is the number of contacts. Finally, the quality metric $\epsilon_f = \min_{\boldsymbol{f}\in\partial\mathcal{W}_f}\|\boldsymbol{f}\|$ is the shortest distance from the origin to the nearest hyper-plane of $\mathcal{W}_f$. Hence, the metric defines a lower bound on the resisted force in all directions.

This concept is easily extended to the torque domain. The reaction torque $\boldsymbol{\tau}_{i,j}$ resulting from a friction cone edge $\boldsymbol{f}_{i,j}$ is $\boldsymbol{\tau}_{i,j} = \boldsymbol{r}_i \times \boldsymbol{f}_{i,j}$, where $\boldsymbol{r}_i$ is a vector pointing from the object's center of mass to the contact point $\boldsymbol{p}_i$. Further, $\mathcal{W}_\tau = \mathrm{ConvexHull}\left(\bigcup_{i=1}^{n}\{\boldsymbol{\tau}_{i,1},\ldots,\boldsymbol{\tau}_{i,m}\}\right)$ is the set of resisted torques. The metric $\epsilon_\tau = \min_{\boldsymbol{\tau}\in\partial\mathcal{W}_\tau}\|\boldsymbol{\tau}\|$ evaluates the grasp's quality by identifying the magnitude of the largest-minimum resisted torque.

### B. Minimum distance to the friction cone

The quality metrics $\epsilon_f$ and $\epsilon_\tau$ analyze the forces that each contact can theoretically exert on the object. However, these metrics do not consider the actual contact forces that the contacts apply to the object. To this end, we define two force-based quality metrics $\delta_{cur}$ and $\delta_{task}$.

Similar to Buss et al. [10], we measure grasp stability in terms of how far the contact forces are from the friction limits. Fig. 2 shows a grasp with the current contact forces $\boldsymbol{f}_{i,cur}$ and the tangential force margins $\bar{\boldsymbol{f}}_{i,cur}$. The vectors $\bar{\boldsymbol{f}}_{i,cur}$ are forces in the tangential direction that point from

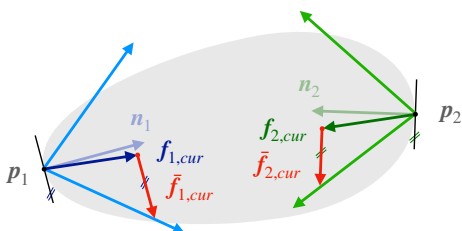

Fig. 2: Grasp with current contact forces $\boldsymbol{f}_{i,cur}$ and tangential force margins $\bar{\boldsymbol{f}}_{i,cur}$ to the friction cones.

$\boldsymbol{f}_{i,cur}$ to the closest point on the friction cone, thereby identifying the direction in which the contact can take the least tangential force before slipping. A grasp with large tangential force margins $\bar{\boldsymbol{f}}_{i,cur}$ is desirable since the contacts are less prone to sliding when an object wrench is applied. Hence, the metric $\delta_{cur}$ measures the average magnitude of the safety margins $\|\bar{\boldsymbol{f}}_{i,cur}\|$ across all contacts $i$.

The set of wrenches that the grasp must resist during task execution (e.g., object weight or wrenches from expected collisions) can often be estimated. Our task-oriented metric $\delta_{task}$ evaluates whether the *current* contact forces of a grasp are suitable to balance the anticipated task wrenches. We calculate the additional contact force $\boldsymbol{f}_{i,add}$ that each contact $i$ must react with to compensate a task wrench $\boldsymbol{w}$ with $\mathbf{G}^+\boldsymbol{w} = (\ \boldsymbol{f}_{1,add}^T\ \ \boldsymbol{f}_{2,add}^T\ \ \cdots\ \ \boldsymbol{f}_{n_c,add}^T\ )^T$, where $\mathbf{G}^+$ is the pseudoinverse of the grasp matrix as defined in [11]. The task contact force is $\boldsymbol{f}_{i,task} = \boldsymbol{f}_{i,cur} + \boldsymbol{f}_{i,add}$ for each contact. Finally, $\delta_{task}$ computes the average magnitude of the tangential force margins $\|\bar{\boldsymbol{f}}_{i,task}\|$ of the task contact forces $\boldsymbol{f}_{i,task}$ to the friction cone.

## III. TACTILE SENSING AND THE REWARD FUNCTION

### A. Train and Test Dataset

Each training sample consists of a tuple $(O, E)$, where $O$ is the object, and $E$ is the wrist pose error sampled uniformly before every episode. There are three object types (cuboid, cylinder, and sphere) with a mass $\in [0.1, 0.4]$ kg and randomly sampled sizes. Fig. 3 visualizes the minimum and maximum object dimensions. The wrist pose error $E$ consists of a translational and a rotational error. We uniformly sample the translational error $(e_x, e_y, e_z)$ from $[-5, 5]$ cm and the rotational error $(e_\xi, e_\eta, e_\zeta)$ from $[-10, 10]$ deg for each variable, respectively.

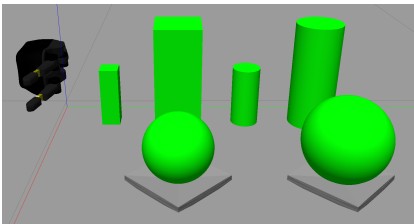

Fig. 3: Minimum and maximum object sizes. We place the spheres on a concave mount to prevent rolling.

We define 8 different wrist error cases for the test dataset. Let $d(a, b, c) = \sqrt{a^2 + b^2 + c^2}$ be the L2 norm of the variables $(a, b, c)$. Table I shows the wrist error cases, where case A corresponds to no error and case H means maximum wrist error. The test dataset consists of 30 random objects $O$ (10 cuboids, 10 cylinders, and 10 spheres). Per object $O$, we randomly generate the eight wrist error cases $\{A, B, \ldots, H\}$ from Table I. Hence, we run $30 \times 8 = 240$ grasping experiments to test one model.

TABLE I: Wrist error cases

| Wrist Error Case | A | B | C | D | E | F | G | H |
|---|---|---|---|---|---|---|---|---|
| $d(e_x, e_y, e_z)$ in cm | 0 | 1 | 2 | 3 | 4 | 5 | 6 | 7 |
| $d(e_\xi, e_\eta, e_\zeta)$ in deg | 0 | 2 | 4 | 6 | 8 | 10 | 12 | 14 |

## A - Initialize World

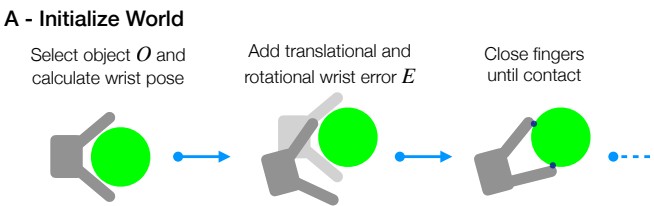

Select object $O$ and calculate wrist pose → Add translational and rotational wrist error $E$ → Close fingers until contact

## B - Grasp Refinement Episode

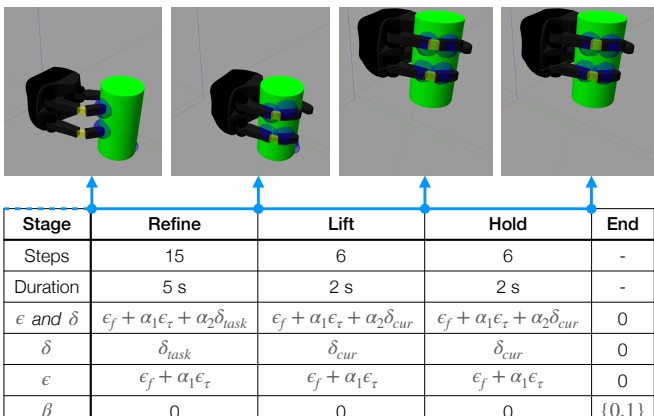

| Stage | Refine | Lift | Hold | End |
|---|---|---|---|---|
| Steps | 15 | 6 | 6 | - |
| Duration | 5 s | 2 s | 2 s | - |
| $\epsilon$ and $\delta$ | $\epsilon_f + \alpha_1\epsilon_\tau + \alpha_2\delta_{task}$ | $\epsilon_f + \alpha_1\epsilon_\tau + \alpha_2\delta_{cur}$ | $\epsilon_f + \alpha_1\epsilon_\tau + \alpha_2\delta_{cur}$ | 0 |
| $\delta$ | $\delta_{task}$ | $\delta_{cur}$ | $\delta_{cur}$ | 0 |
| $\epsilon$ | $\epsilon_f + \alpha_1\epsilon_\tau$ | $\epsilon_f + \alpha_1\epsilon_\tau$ | $\epsilon_f + \alpha_1\epsilon_\tau$ | 0 |
| $\beta$ | 0 | 0 | 0 | {0,1} |

Fig. 4: Overview of one algorithm episode. (A) Initialization of hand and object. (B) We split the grasp refinement algorithm into four stages and compare four reward frameworks: (1) $\epsilon$ and $\delta$, (2) only $\delta$, (3) only $\epsilon$ and (4) the non-tactile binary reward baseline $\beta$. The weighting factors of $\alpha_1 = 5$ and $\alpha_2 = 0.5$ were empirically determined.

### B. State and Action Space

The state vector $s$ consists of 7 joint positions (1 finger separation, 3 proximal bending, 3 distal bending degrees of freedom), and 7 contact cues (3 on proximal links, 3 on distal links, and 1 on palm) that include contact position, contact normal and contact force, which have 3 $(x, y, z)$ components each. The dimension of the state vector is $s \in \mathbb{R}^{7+7\times(3\times3)=70}$. Note that we do not assume any information about the object (e.g., object pose, geometry, or mass) in the state vector. The contact normals and positions are provided in the wrist frame, while the contact forces are represented in the contact frame. The action vector $a$ consists of 3 finger position increments, 3 wrist position increments and 3 wrist rotation increments. The action vector's dimension is $a \in \mathbb{R}^{3+3+3=9}$. The policy $\pi_\theta$ is parametrized by a neural network with weights $\theta$. The network is a multi-layer perceptron with four layers (70, 256, 256, 9). We use the stable-baselines3 [12] implementation of the soft actor-critic (SAC) [13] algorithm and train for 25000 steps.

### C. Experimental Setup

We simulate the three-fingered ReFlex TakkTile hand (RightHand Robotics, Somerville, MA USA) using a custom Gazebo [14] simulation environment and the DART [15] physics engine. We model the under-actuated distal flexure [16] as a rigid link with two revolute joints (one between the proximal and one between the distal finger link). Further, we approximate the finger geometries as cuboids to

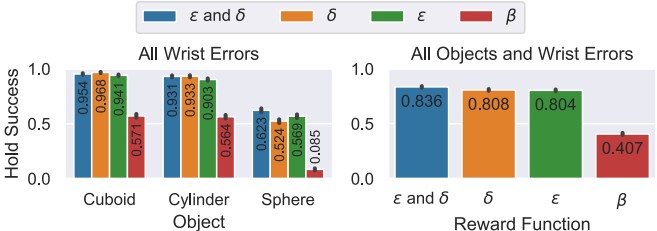

Fig. 5: Test results for reward frameworks.

reduce computational load. Our source code is available at github.com/axkoenig/grasp_refinement.

Fig. 4 shows an overview of one training episode. In stage (A), we initialize the world. Thereby, we randomly generate a new object, wrist error tuple $(O, E)$ (or we select one from the test dataset). We assume a computer vision system and a grasp planner that produces a side-ways facing grasp at a fixed 5 cm offset from the object's center of mass. We add the wrist pose error $E$ to this grasp pose to simulate calibration errors and close the fingers of the robotic hand in the erroneous wrist pose until the fingers make contact with the object. Consequently, the grasp refinement episode (B) starts. We divide each episode into three stages, as displayed in Fig. 4. Firstly, the policy $\pi_\theta$ *refines* the grasp. Afterward, the agent *lifts* the object by 15 cm via hard-coded increments to the wrist's $z$-position and *holds* the object in place to test the grasp's stability. The policy $\pi_\theta$ can update the wrist and finger positions while lifting and holding. The control frequency of the policy in all stages is 3 Hz, while the update frequency of the low-level proportional–derivative (PD) controllers in the wrist and the fingers is 100 Hz.

As shown in the table of Fig. 4, we use the analytic grasp stability metrics from section II as reward functions. We compare the following reward configurations: (1) both $\epsilon$ and $\delta$, (2) only $\epsilon$, (3) only $\delta$ and (4) the baseline $\beta$. Fig. 4 shows that $\delta$ refers to $\delta_{task}$ in the *refine* stage to measure expected grasp stability before lifting and $\delta_{cur}$ in the *lift* and *hold* stages to measure current stability. Further, $\epsilon$ is a weighted combination of $\epsilon_f$ and $\epsilon_\tau$. While $\epsilon$ *and* $\delta$, $\delta$, and $\epsilon$ provide stability feedback after every algorithm step, the baseline $\beta$ gives a sparse reward after the holding stage, indicating if the object is still in the hand (1) or not (0).

### D. Results and Discussion

For all experiments in this paper, we average over 40 models trained with different seeds for each framework. The error bars in all plots represent ±2 standard errors. Fig. 5 summarizes the performance on the test dataset. Our main observation is that combining the geometric grasp stability metric $\epsilon$ with the force-agnostic metric $\delta$ yields the highest average success rates of 83.6% across all objects (95.4% for cuboids, 93.1% for cylinders, and 62.3% for spheres) over all wrist errors. The $\epsilon$ *and* $\delta$ framework outperforms the binary reward framework $\beta$ by 42.9%. The p-values for our results $\mu_{\epsilon\,and\,\delta} > \mu_\delta$, $\mu_{\epsilon\,and\,\delta} > \mu_\epsilon$ and $\mu_{\epsilon\,and\,\delta} > \mu_\beta$ (where $\mu_x$ is the mean performance of framework $x$) are all $\ll 0.001$ and are hence statistically significant. We also notice that

the combination between $\epsilon$ *and* $\delta$ is particularly helpful for spheres. The average performance of all frameworks on spheres is greatly reduced, while the algorithms trained with $\beta$ especially struggle to grasp spheres.

This study investigates the tactile sensing needs in the reward of RL grasping controllers by incorporating highly accurate contact information via analytic grasp stability metrics. The results demonstrate that information about contact positions and normals encoded in $\epsilon$ combines well with the force-based information in the $\delta$ reward. This result motivates building physical robotic hands capable of sensing these types of information. The low success rates for the spheres may be because they can roll and are therefore harder to grasp (cuboids and cylinders move comparatively less when touched by fingers or the palm). The $\beta$ framework performs worst after the defined number of training steps, which is unsurprising because shaped rewards are known to be more sample efficient than sparse rewards [17].

## IV. Tactile Sensing and the State Vector

### A. Experimental Setup

In a second experiment, we investigate the effect of contact sensing resolution in the state vector on grasp refinement. We compare four contact sensing frameworks. The *full* contact sensing framework receives the same state vector $s \in \mathbb{R}^{70}$ as in section III-B. In the *normal* framework, we only provide the algorithm with the contact normal forces and omit the tangential forces ($s \in \mathbb{R}^{56}$). In the *binary* framework we only give a binary signal whether a link is in contact (1) or not (0) ($s \in \mathbb{R}^{56}$). Finally, we solely provide the joint positions in the *none* framework ($s \in \mathbb{R}^{7}$). We adjust the size of the input layer of the neural network from section III-B to match the size of the state vector of each framework. The reward function in these experiments is $\epsilon$ *and* $\delta$ from Fig. 4. Hence, all contact sensing frameworks receive contact information indirectly via the reward.

### B. Results and Discussion

In Fig. 6, we observe that the frameworks which receive contact feedback (*full*, *normal*, *binary*) outperform the *none* framework by 6.3%, 6.6% and 3.7%, respectively. Providing *normal* force information yields a performance increase of 2.9% compared to the *binary* framework. However, training with the *full* contact force vectors only increases the performance by 2.6% compared to the *binary* framework. As expected, performance decreases for larger wrist errors. The results $\mu_{normal} > \mu_{binary}$ and $\mu_{normal} > \mu_{none}$ are statistically significant (p-values $\ll 0.001$), while the result $\mu_{normal} > \mu_{full}$ is not (p-value 0.2232).

This experiment studies how contact sensing resolution in the policy's state vector is related to grasp success when training with fully contact informed rewards. Thereby, we investigate the viability of our hypothesized training and deployment workflow in Fig. 1. The improvements for the *normal* force framework over the *binary* and *none* frameworks are small. The results suggest that an affordable *binary* contact sensor suite, or even no contact sensing at all, may be

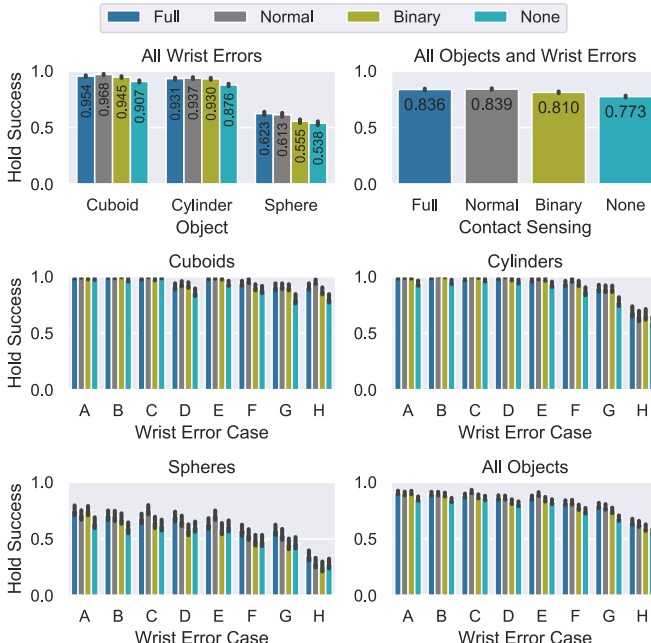

Fig. 6: Test results for contact sensing frameworks.

suitable if a small decrease in performance is tolerable. This result supports our hypothesis that RL grasping algorithms are deployable to hands with reduced contact sensor resolution at little performance decrease when incorporating rich tactile feedback at train time. The algorithms trained with the *full* force vector perform approximately on par with the ones that receive the *normal* force information. This could be due to three reasons. (1) The *full* force framework has the most network parameters and requires even longer training times. (2) The model fails to represent the concept of the friction cone internally. An alternative representation of the tangential forces could be a solution (e.g., providing a margin to the friction cone instead of a tangential force vector). (3) Simulated contact forces are prone to instability [18], especially when simulating robotic grasping [19].

## V. Conclusion

This paper investigated the importance of tactile signals in the reward and the policy's state vector to identify the tactile sensing needs in RL-based grasping algorithms. We found that rewards incorporating contact positions, normals, and forces are the most powerful optimization objectives for RL grasp refinement controllers. While this tactile information is essential in the reward function, we uncovered that reducing contact sensor resolution in the policy's state vector decreases algorithm performance only by a small amount. This result has implications for the design of physical grippers and their training and deployment workflows.

In future work, we aim to build physical robotic hands with advanced sensing capabilities to calculate grasp metrics. Secondly, we want to test the proposed training and deployment workflow, providing only limited contact information in the state vector and testing the algorithm on other robotic hands.

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
