# OpenReview forum: "Tactile Sensing and its Role in Learning and Deploying Robotic Grasping Controllers"
_ICRA.org/2022/Workshop/Contact-Rich — ICRA 2022 Workshop: RL for Manipulation Poster_

### Official Review · Reviewer_2Ffx · 2022-05-05
**Interesting paper that could benefit from re-structuring**

**Rating:** 6
**Confidence:** 4

**Review:**

This paper investigates the effects of incorporating analytic grasp metrics into the reward function and state of RL-based grasping algorithms. The authors employ a combination of previously defined metrics ($\epsilon$) and further introduce their own ($\delta$).
In their experiments regarding the incorporation of both metrics in the reward function, the authors find that the combination of the metrics brings higher grasp success results compared to a non-tactile baseline. In addition, the performance degradation if using only one of the metrics is not major, leading to the conclusion that since the reward function is not crucial for the deployment of the algorithm, it is possible to train using complicated sensors and deploy using simpler and cheaper ones. Furthermore, by incorporating the metrics in the state space of the agent, they conclude that the performance improvement when using normal forces VS a binary indication of contact is not substantial enough to justify always using the more complicated sensing.
The results of this work are definitely relevant to the community and could be interesting for a plethora of use cases. The paper itself is well-written for the most part, although I think that a few modifications and addressing some of the following comments could improve both readability and future work contributions.

1. In the introduction you mention that the reward signal is not important for the testing phase. This is technically correct when considering a fully deployed algorithm but especially in RL settings, an indication of the test performance continues to be important to evaluate the final policy. Furthermore, I believe that motivating the usage of the reward VS state should be a little more thorough for future work. For example, in several cases having this information in the state allows you to react to potentially problematic cases by simply checking it against constraints, while in the reward it is being weighed together with more information and cannot be used as such an indicator.

2. I would urge you to consider renaming the $\delta_{cur}$ to $\delta_{margin}$ or something along those lines, as $\delta_{task}$ is referred to as "a metric evaluating the $\textit{current}$ contact forces" which is confusing to the reader.

3. Section III could benefit from restructuring. For example, you mention technical details such as how the wrist pose error is sampled, before you define what it is. Maybe III.C should be before III.A to give the reader some context. You could also start this section with a little preface of what is to come i.e. what your experiments are going to prove.

4. Try to refer to your metrics in the same way throughout the paper. E.g. in section III.C. you refer to $\delta_{task}$ as the expected grasp stability and to $\delta_{cur}$ as the current stability.

5. In section III.D, could you please clarify how $\delta$ is force agnostic since by definition both of the deltas are related to forces?

6. In section III.D you also mention that the low success for spheres is potentially because they more more when touched by the fingers. I believe that in future work it would be nice to support this claim with hard numbers, e.g. report the average displacements/rotations of the objects during manipulation and maybe even what forces they were accompanied by.

7. Given the control frequency for the policy updates is at 3 Hz, is it possible to utilize your approach for events like slippage detection?

8. Lastly, in the future it would be very interesting to see a more in-depth commentary of the results reported in section IV.B regarding the relatively small improvement between "none" and the other baselines.

In conclusion, I think this paper is definitely relevant to the community and a few modifications could greatly improve it.

---

### Official Review · Reviewer_nyBF · 2022-05-09
**Great work for introducing tactile signals as RL reward for grasp refinement, and analyzing the performance of reduced tactile resolution**

**Rating:** 7
**Confidence:** 3

**Review:**

Summary: This work introduced to add rich contact reward to improve the RL training for grasping refinement. Two grasp stability metrics were applied to provide a dense reward. The experiments showed the reward based on tactile grasp stability outperforms the baseline where only a sparse reward is provided. The ablation study compared the variations with different tactile resolutions, showing the possibility of deployment of low-cost tactile while training with rich tactile reward.

Pros:
- The paper is clear and well-written.
- The idea of incorporating tactile signal as grasp stability for RL reward is novel, and the experiments show its effectiveness compared to the previous sparse reward.
- The experiments analyze the performance of the different tactile resolutions. It is helpful for actual hardware design and deployment in the future.

Cons:
- The studied object set is relatively simple. It would to interesting to see the generalization to more complex objects, like the YCB dataset.
- Since the real-world data would be noisier. It would help to see the performance with noisier tactile signals and discuss the possible sim2real gap.